# The Research Field of Meat Preservation: A Scientometric and Visualization Analysis Based on the Web of Science

**DOI:** 10.3390/foods12234239

**Published:** 2023-11-24

**Authors:** Jingjing Zhang, Zixiang Wei, Ting Lu, Xingzhen Qi, Lan Xie, Silvia Vincenzetti, Paolo Polidori, Lanjie Li, Guiqin Liu

**Affiliations:** 1Shandong Engineering Technology Research Center for Efficient Breeding and Ecological Feeding of Black Donkey, College of Agronomy and Agricultural Engineering, Liaocheng University, Liaocheng 252000, China; jingjing.zhang@unicam.it (J.Z.); 19993443992@163.com (T.L.); qxz17354603787@126.com (X.Q.); m13210458205@163.com (L.X.); 2School of Biosciences and Veterinary Medicine, University of Camerino, Via Circonvallazione 93, 62024 Matelica, MC, Italy; silvia.vincenzetti@unicam.it; 3Key Laboratory of Industrial Fermentation Microbiology, College of Biotechnology, Tianjin University of Science and Technology, Tianjin 300222, China; 17637506118@163.com; 4School of Pharmacy, University of Camerino, Via Gentile da Varano, 62032 Camerino, MC, Italy; paolo.polidori@unicam.it; 5Office of International Programs, Liaocheng University, Liaocheng 252000, China

**Keywords:** meat preservation, bibliometrics, knowledge structure, CiteSpace, VOSviewer, Web of Science

## Abstract

Meat plays a significant role in human diets, providing a rich source of high-quality protein. With advancements in technology, research in the field of meat preservation has been undergoing dynamic evolution. To gain insights into the development of this discipline, the study conducted an analysis and knowledge structure mapping of 1672 papers related to meat preservation research within the Web of Science Core Collection (WOSCC) spanning from 2001 to 2023. And using software tools such as VOSviewer 1.6.18 and CiteSpace 5.8.R3c allowed for the convenient analysis of the literature by strictly following the software operation manuals. Moreover, the knowledge structure of research in the field of meat preservation was synthesized within the framework of “basic research—technological application—integration of technology with fundamental research,” aligning with the research content. Co-cited literature analysis indicated that meat preservation research could be further categorized into seven collections, as well as highlighting the prominent role of the antibacterial and antioxidant properties of plant essential oils in ongoing research. Subsequently, the future research direction and focus of the meat preservation field were predicted and prospected. The findings of this study could offer valuable assistance to researchers in swiftly comprehending the discipline’s development and identifying prominent research areas, thus providing valuable guidance for shaping research topics.

## 1. Introduction

The desire to offer meat products characterized by superior quality, affordability, eco-friendliness, and appeal to consumers has led meat processors, distributors, retailers, and industrialists to strive for the provision of such desired attributes [1]. Meat stands as a crucial source of animal-based nourishment for humans, offering a rich supply of high-quality protein. In recent years, the consumer demand for meat has surged significantly, driven by its appealing taste and nutritional benefits [2]. According to Ritchie (2017), per capita meat consumption has significantly increased by approximately 20 kg since 1961, reaching 43 kg per capita in 2014 [3]. Nonetheless, while meat is a rich source of protein and fat, it is inherently vulnerable to spoilage during processing, transportation, and sales. This susceptibility could contribute to environmental problems, economic losses, and poses significant health risks to consumers. For example, chemical additives used to prevent meat spoilage can enter water sources with wastewater, leading to the pollution of aquatic environments [4]. Troy et al. (2016) reported that food preservation and processing are energy-intensive processes; therefore, limiting energy usage can not only reduce environmental impact (e.g., greenhouse effect) but also decrease economic losses [5,6]. Thus, the focal point in meat preservation has revolved around the examination of spoilage and deterioration mechanisms, encompassing digestive enzymes, microorganisms, as well as fat and protein oxidation. Additionally, the application of various preservation techniques has been paramount in efforts to extend the shelf life of meat products, with the aim of meeting the growing demand for high-quality and safe meat products.

The high protein content and abundant moisture and fat in meat make it highly susceptible to microbial proliferation and natural oxidation, leading to spoilage [7]. Consequently, finding solutions to maintain meat quality and sensory attributes has been a focal point for professionals and researchers in the meat industry. Common meat preservation techniques primarily fall into three categories: (a) temperature and gas control [8]. These techniques, including freezing, chilling, and superchilling, are designed to slow or limit the rate of meat spoilage by controlling the temperature below the optimal range to inhibit the growth of microorganisms. (b) The use of chemical preservatives, biological preservatives, and physical treatments [9,10]. Preservatives are the substances used to extend the shelf life of meat by reducing microbial proliferation, including chlorides, nitrites, sulfides, and organic acids. The addition of these antimicrobial preservatives during processing can be combined with refrigeration techniques to provide enhanced protection for meat products. (c) Moisture content control in meat [11,12]. The water activity in meat products directly affects the microbiological safety of food and can be controlled through methods such as drying, refrigeration, the addition of chemicals, or their combinations. For instance, chemical substances like sodium chloride and sugar are typically used to regulate water activity by binding free water, thereby inducing osmotic imbalances, and ultimately inhibiting cell growth. With the advancement of meat preservation technology and the promotion of the clean label concept, safe, minimally toxic, green, and efficient chemical and natural antimicrobial agents, as well as non-thermal preservation techniques, have gradually replaced traditional meat preservation methods such as hot water treatment, chlorinated surface rinsing, freezing, air drying, and fermentation [13,14,15]. However, the term “clean label” has not yet been clearly defined by food authorities and is quite subjective. In general, the term refers to products that do not contain or use as few “artificial” additives as possible and are produced “naturally” or based on traditional formulation methods already known to consumers. The use of “clean labels” in meat products helps to meet consumers’ desire for “natural, nutritious and healthy” properties in food [16,17].

Scientific literature can be used to help researchers understand the structure and dynamics of a research field. However, traditional meat research reviews cannot satisfy people’s comprehensive understanding of this field. Furthermore, obtaining a large-scale literature review from a huge corpus of relevant research papers is not easy for researchers [18]. As subject areas progress, the ability to present the dynamic visual knowledge within the field of meat preservation research is becoming increasingly vital. Bibliometrics relies on statistical and visualization techniques, is used to summarize historical research results and predict future research trends based on scientific literature databases, and is extensively employed in a range of fields, including but not limited to food, health, environmental protection, and agricultural management [19,20,21]. In a bibliometric analysis conducted by de Melo, A. M (2021), it was determined that the bacupari fruit holds substantial potential for application in the realms of both food and health [22]. Notably, the amalgamation of traditional review papers, which offer comprehensive insights into the evolution of a research discipline, empowers researchers to attain a more precise, clear, and in-depth understanding of the research field, facilitating a swift grasp of the research frontiers. At the same time, the widespread adoption of bibliometric software, such as VOSviewer, CiteSpace, and BibExcel, aids researchers in effortlessly mapping the knowledge networks within subject areas, allowing them to discern research structures and hotspots [23,24,25]. Therefore, it seems particularly important to conduct a specific bibliometric analysis of meat preservation research.

This study used bibliometrics to scrutinize the developmental structure of the meat preservation field utilizing the WOS database, with the aim being to gain insight into future research trends in this field. The results of this study will help scientists to better understand the research progress and relevant research hotspots in the field of meat preservation, as well as providing reference points for future research work in this field.

## 2. Materials and Methods

### 2.1. Data Sources and Search Strategy

The articles used in the present study for analysis were selected from the Web of Science Core Collection. Search date: 1 January 2001/31 August 2023. Search formula: TS = (“preserva*”) OR TS = (“extend*shelf life”) OR TS = (“extend* storage”) AND TS = (“meat”). Document Type: Articles or Review Articles or Early Access. A total of 2311 articles were initially retrieved through the search. Subsequently, literature records from the WOS database were exported in full-record format. Duplicate documents were removed using the “Remove Duplicates” function in the Citespace software 5.8.R3c. Following this, based on comprehensive reviews of the full texts, irrelevant studies were excluded. All authors assessed their relevance until a consensus was reached. In the end, 1672 articles meeting the criteria were obtained for further analysis.

### 2.2. Software Used

For the visual analysis in this study, Citespace 5.8.R3c and VOSviewer 1.6.18 were employed. Origin 2022 and Scimago Graphica 1.0.17 were used to create visual representations, while data statistics were conducted using Microsoft Excel 2019. Within the scope of this study, various plots and visual representations were created using different software tools. For example, origin software 2022 was employed for plotting the annual publication volume and citation trends. Citespace software was used to analyze and generate the journal double citation map, co-citation journal clustering map, and co-citation journal timeline map. For specific drawing methods, see the research of Citespace developer Chen [26,27]. VOSviewer aided in creating the keyword spatio-temporal map and keyword clustering hotspot map [28]. The network visualization map of inter-country cooperation was jointly analyzed and generated using VOSviewer and Scimago Graphica. The specific procedures involved regions belonging to the same country were merged by VOSviewer (e.g., Taiwan, Hong Kong, and Macau were represented by the People’s Republic of China; Northern Ireland, Scotland, and Wales were represented by England), and Scimago Graphica was used to create the network visualization map. Furthermore, the rest of the tables were drawn using Excel statistics.

## 3. Results and Discussion

### 3.1. The Field of Meat Preservation Research

The field of meat preservation research is focused on studying how to prolong and protect the quality and food safety of meat products during storage and transportation. It is a complex research area that involves multiple disciplines working together. Moreover, it is crucial for researchers to dynamically and intuitively analyze and assess the variations in research between different disciplines within this research field. However, this issue can be effectively addressed by utilizing double-layered thematic overlays on a global scientific map and examining the characteristics of publication combinations from both spatial and temporal perspectives [26]. Therefore, we used Citespace to generate a double map of meat preservation research.

The dual map of meat preservation research mainly establishes the relationship between the citing literature and the cited literature using journals from the WOS database [29]. The left half of the figure represents the current status of meat preservation research with the subject distribution of citations, while the right half represents the research foundation of meat preservation studies based on the subjects of citations. The wavy curve connects the relationship between the research status and the research foundation. And the size of the ellipse reflects the number of papers published in each discipline; meanwhile, the bar chart at the top illustrates the yearly variations in the number of published papers [30]. In the dual map, the connection between published articles in the meat preservation research field and the disciplines of the cited articles is depicted for the period 2001–2022. Similar disciplines are denoted with purple labels, and the various research subjects related to meat preservation are organized into distinct color-coded sections from top to bottom in Figure 1. The purple areas cluster within the disciplines of physics, materials, and chemistry. The blue sections are associated with the disciplines of ecology, earth, and marine studies. Additionally, the orange regions are connected to the disciplines of molecular biology and immunology, while the green areas pertain to dentistry, dermatology, surgery, and various medical and clinical fields. Moreover, the primary yellow citation paths in Figure 1 reveal that the literature published in the veterinary and animal sciences disciplines were predominantly cited in the fields of plant ecology, zoology, environmental science, toxicology, and nutrition.

The red dotted line in the enlarged figures at the top of the dual map diagram represents the dynamic changes in the disciplines of citing and cited journals since 2001. The published literature existed as part of the veterinary/animal/science discipline from beginning to end, and the current research hotspot has been concentrated in the discipline of veterinary/animal/science. Concurrently, the disciplines of the cited journals fluctuated repeatedly from 2001 to 2022, but the subject remains in the discipline of veterinary/animal/parasitology, which was almost consistent with the dynamic change in the citation discipline. This suggests that both the citing literature and the cited literature had a certain degree of stability.

### 3.2. Annual Volume of Articles and Citation Trend

The volume of periodical paper publications serves as a significant indicator for assessing the research status of an industry [31]. An annual analysis of 1672 journal papers in the field of meat preservation from the WOS database is presented in Figure 2. Overall, the analysis of publication numbers and citations reveals certain trends. Notably, from 2001 to 2009, there were relatively few studies on meat preservation, with only 216 articles published, accounting for a mere 15.5% of the total. This period signifies the infancy of relevant research in the field.

However, a significant uptick in the number of related articles occurred between 2010 and 2022, indicating a sustained increase in the popularity and attention to this research area, culminating in the development of a notable research scale. Furthermore, it is important to note that in 2023, only 129 articles were published, which is based solely on the number of periodical papers from the first three months of the year.

The annual citation count of the papers exhibited a pattern similar to the number of articles published. Notably, the citation frequency in 2020 and 2022 surpassed 5000, indicating the sustained interest of researchers in this field and a preference for citing newly published literature.

### 3.3. Meat Preservation Research among Countries and Institutions

The number of publications and amount of international cooperation among countries are illustrated in Figure 3. In this visual representation, each circle represents a specific country, with the size of the circle corresponding to the number of publications from that country [32]. Additionally, the connections between circles denote cooperative relationships between the respective countries.

The thickness of each line signifies the degree and quantity of collaboration between countries, while the depth of color indicates the overall intensity of their cooperation [33]. Obviously, China and Spain had a high intensity of cooperation, among which Spain had close cooperative relations with Italy, Brazil, and Australia. Furthermore, China and the United States also maintained a closely collaborative relationship. When considering the data presented in Table 1, it becomes evident that China published 238 articles, accounting for 17.02% of the total. Spain contributed 153 articles, representing 10.94% of the total publications, while the United States published 124 articles, making up 8.87% of the total. And the remaining articles published were from countries such as Brazil, India, Iran, Italy, South Korea, Turkey, and Australia.

In summary, China and Spain not only lead in terms of research output in the field of meat research but also maintain positive collaborations with numerous other countries. Interestingly, it is noteworthy that half of the top 10 countries involved are developing countries, which could be attributed to the significant role that the meat industry plays in the economies of these developing countries. 

Subsequently, we made statistics on the top 10 journals in terms of publication volume in Table 2. The journal of Meat Sci. topped the list, followed by J. Food Process Pres. and LWT-Food Sci. Technol. The top 10 journals are all in the fields of Food Science and Technology, which also provides support for subsequent researchers to analyze research preamble and submit articles.

### 3.4. Keywords in Meat Preservation Research

Keywords provide an overview of the core content of an article and can be used to analyze the objects and content of the research. Visual analysis of keyword clustering and recurrence frequency allows researchers to quickly grasp the main directions and hot topics of the research [27]. Additionally, keyword co-occurrence analysis could help assess the relevance and impact of the literature, supporting researchers in evaluating and formulating research objectives. In the course of the research, a keyword co-occurrence analysis was carried out within the field of meat preservation using VOSviewer software. A total of 324 keywords were extracted, utilizing a co-occurrence threshold of more than 10 times. Moreover, synonyms were amalgamated, resulting in 262 distinct keywords that were used to draw the cluster density map.

As depicted in Figure 4, all the keywords are effectively grouped into five clusters, primarily categorized under meat variety, meat physical and chemical quality, research on meat quality, preservation technology, and related subjects. However, we observed that there were no clear boundaries between the clusters, indicating close connections between them. This finding suggested there was significant overlap between different research directions. For instance, keywords such as “essential oil, packaging, nisin” and “sensory attributes, microorganisms, temperature”, which are located in different clusters, intersect with each other. This reminds us that the impact of microorganisms and the environment on meat quality and the combination of preservation technologies may currently be the hotspot and focus of research. Further analysis of co-citation in the references is needed to address the aforementioned issues.

In Cluster 1, the keywords primarily revolve around research related to microorganisms, meat quality, and safety, with representative terms such as listeria-monocytogenes, lactic acid bacteria, and spoilage. Cluster 2 emphasizes research on meat antioxidants, featuring keywords like antioxidant activities, lipid oxidation, and natural antimicrobials. Cluster 3 is focused on various aspects of meat quality and shelf-life studies, with representative keywords like meat, shelf life, and quality. In Cluster 4, the keywords center around research on meat antimicrobials, including terms like antibacterial activity, essential oils, and antibacterial agents.

Lastly, Cluster 5 is dedicated to the study of preservation technology, featuring representative keywords such as modified atmosphere packaging, nisin, and packaging techniques.

We then created a density network diagram of the keywords, as seen in Figure 5, where the color gradient from left to right signifies the evolving emphasis on these keywords over time. Blue represents earlier research in this field, while yellow indicates the current research frontier [34]. In the field of meat research, a large number of detailed and in-depth research has been carried out on the physical and chemical properties of meat, as well as on microbial indicators and representative microorganisms in meat. Furthermore, by observing the size of the circles representing keywords, it was found that the current research hotspots include lipid oxidation, active packaging, antibacterial activity, essential oil preservation, film preservation, and so on. For example, Borzi et al. (2019) conducted a study in which they prepared antioxidant polyamide active packaging films through leaching. They observed that these active packaging films effectively prolonged the storage period of pork, extending it up to 23 days, while also significantly reducing fat oxidation [35]. Xavier et al. (2021) prepared cinnamon essential oil into nano-microcapsules and then combined it with corn alcohol solution to make active packaging for beef preservation, which effectively extended the shelf life of the beef and delayed fat oxidation [36].

### 3.5. Co-Citation in Meat Preservation Research

The co-citation analysis of the literature stands as a crucial component of citation analysis. It involves selecting high-impact literature within a particular subject area and examining the simultaneous citations of these works as a means to gauge the similarity of their respective research areas. This serves as a foundational approach for studying the trends in the development of the discipline [24]. In CiteSpace, the parameters are configured, and the time frame is established based on the literature’s predetermined years spanning from 2001 to 2022. Each year is segmented into distinct time slices, with the node type designated as “literature co-citation”. Simultaneously, the top 100 most frequently cited pieces of literature are extracted for analysis. Then, a literature co-citation network graph is generated. Figure 6 displays the network clustering map of co-cited literature published between 2001 and 2023, presenting a total of 17 citation clustering color blocks.

It is important to note that a lower number of clustering color blocks indicates a higher number of papers within the set of clusters being represented. The clustering module’s Q value exceeded 0.8687, indicating a significant distinction in the clustered cited literature. Additionally, each clustering color block achieved an S value greater than 0.9, affirming the credibility and robustness of the clustering results [37].

We selected the top 10 co-cited literature clusters, which were used to make a timeline. In this visualization, the length of the solid line within each row signifies the duration of the study, while the size of the circle denotes the current point in time when the cited literature is highlighted. Notably, the longest duration span was observed in Figure 7 for Clusters 1 and 3, which have remained active for 11 years. On the other hand, Clusters 7, 8, and 9 have not displayed significant recent activity. Clusters 0, 1, 3, and 10 have demonstrated sustained activity in recent years, as confirmed by the average years presented in Table 3 and Figure 6. Following this, we tallied the highly referenced and cited papers within each cluster based on the labels provided in Table 3. We then conducted an analysis, with the cited literature representing the knowledge base and the citing literature representing the research frontier [38]. The clustering labels are further organized based on their research categories and the associated technologies. Notably, Cluster 0 and Cluster 3 exemplify well-established advancements in the field of meat preservation, emphasizing strategies for extending shelf life through synergistic enhancements. The earliest cited literature in both collections is the article by Fernandez-Pan I, published in the journal J SCI FOOD AGR in 2013, which explores the enhancement of chicken shelf life through the synergistic use of oregano and clove essential oils [39]. Additionally, the literature with burst citations in both collections is predominantly concentrated within the period of 2014 to 2018.

Cluster 1 and Cluster 7 contain fundamental research in the field of meat preservation. The related studies have highlighted that the primary cause of meat spoilage is attributed to microorganisms [48]. These microorganisms contribute to food safety concerns within meat products by breaking down proteins, fats, and other components, ultimately producing volatile amines and waste materials. The microbial degradation of proteins and fats results in the production of volatile amines and fecal odors, which in turn contribute to food safety concerns in meat products. Thus, the adoption of targeted strategies to manage the dominant spoilage bacteria can effectively prolong the shelf life of meat. Annalisa Casaburi (2015) reviewed meat spoilage microorganisms, including Fusobacterium and Pseudomonas [49]. The research within the collection is divided into two distinct periods: the first spanning from 1997 to 2003, and the second from 2010 to 2020. This division signifies the continuous investigation of microorganisms in meat throughout the research phase. The shift between these two periods is primarily driven by the replacement of tradition methods of in-depth microorganism research, including culture methods, with the high-throughput sequencing and newly developed molecular sequencing techniques. Furthermore, it is essential to highlight that preservation strategies should be thoughtfully adapted to account for the evolving dynamics of the dominant spoilage bacteria across different stages.

In the realm of meat preservation applications, nanoemulsion is frequently utilized to encapsulate and safeguard essential oils, aimed at diminishing their volatility and hydrophobic characteristics, particularly when these essential oils possess antimicrobial properties. Therefore, the authors have consolidated Cluster 2 and Cluster 5 into a collective entity, characterized by its continuous activity. A large number of studies have pointed out the efficacy of transparent or translucent emulsions with diameters ranging from 100 to 600 nm in enhancing both the quality and shelf life of meat products [53,57,86]. For instance, Xiong (2020) developed nanoemulsions containing oregano essential oil and resveratrol in a piece of highly co-cited research. These emulsions were subsequently applied to pectin films, effectively extending the freshness of pork by 3–4 days [55]. Extending the shelf life of meat products through nanoemulsion combined with coating loading has emerged as a popular and valuable research direction.

Cluster 4 and Cluster 9, along with Cluster 13 and Cluster 14, have been amalgamated into a distinct category of research directions aimed at extending the shelf life of meat. These groupings encompass physical techniques designed to protect meat quality and prolong shelf life using methods like lamination and air conditioning preservation. A multitude of studies in the field of meat preservation have underscored the efficacy of substrates such as chitosan, sodium alginate, and konjac gum in the form of laminating films with antibacterial properties [87]. The utilization of air conditioning packaging in meat preservation research has demonstrated its efficacy in effectively extending shelf life and retarding color loss. However, it has been suggested that air conditioning can also exert a notable influence on the proliferation of parthenogenic and anaerobic bacteria during the storage of meat products. This can lead to the rapid growth of bacteria, including lactic acid bacteria, even when the color of the meat has not significantly deteriorated. Simultaneously, consumers harbor negative perceptions of carbon monoxide (CO), viewing it as a potentially hazardous gas [88]. Addressing how to mitigate the negative issues associated with gas preservation is a matter for researchers to contemplate. Additionally, it is necessary to develop films that are non-toxic, non-hazardous, readily degradable, and economically cost-effective, all while maintaining high barrier properties.

The authors have concluded that Cluster 6 and Cluster 16 represent containment and inclusion relationships. Within these clusters, common antimicrobial agents are categorized into chemical preservatives and biological preservatives, of which secondary metabolites produced through bacterial metabolism, such as nisin, laminin, and ε-polylysine, have a well-established history of utilization in food and meat industry production. Research has indicated that the use of bacteriocins can effectively inhibit bacterial proliferation, thus extending shelf life in a noteworthy manner [89,90].

Cluster 8, Cluster 11, and Cluster 12 served as the classification subsets for the subjects under investigation. Notably, the timeline graph indicates that the study of rabbit meat has been emerging, signifying a growing trend of researching meat with distinctive local characteristics in recent years.

Likewise, Clusters 10 and 15 have brought together fundamental research in the field of meat preservation. Within these clusters, it has been established that meat protein oxidation and lipid oxidation can result in undesirable off-flavors, color degradation, and alterations in textural properties [18,91,92]. Employing physical or chemical methods to reduce fat and protein oxidation and peroxidation can effectively enhance quality indicators.

Citation bursts are indicative of the depth of engagement in a specific field over a particular time period. Therefore, the top 10 bursts of cited literature in the realm of meat research are listed in Table 4. Notably, five of the highly emergent cited literatures are review articles, all of which have received significant attention. This underscores the widespread appreciation of reviews among researchers, and their role in effectively portraying the research foundation and advancement of current topics. Subsequently, we counted the literatures that continue to exhibit burst citations to the present day, signifying the prevailing research focus that is detailed in Table 5. Notably, the topics covered in the cited literature predominantly revolve around plant extracts and plant essential oils, indicating that the pursuit of natural and effective biopreservatives derived from plants has emerged as a prominent and current research hotspot.

### 3.6. Knowledge Structure Framework in the Field of Meat Preservation

Based on the preceding analysis of the evolution of research disciplines in meat storage and preservation, it becomes evident that these disciplines evolve along a trajectory characterized by “basic research—technology application—technology application combined with basic research”. This developmental pathway is further illustrated in Figure 8, which outlines three distinct knowledge areas. The research within the domain of meat storage and preservation is divided into these three knowledge areas, as depicted in Figure 8, with each area being the focus of ongoing investigation.

First of all, the green segment signifies the realm of basic research. This research involves the examination of various types of meat, and it entails the establishment of shelf-life criteria. These criteria are determined through an analysis of changes in the population of total bacteria and conditionally pathogenic bacteria, alongside the evaluation of physicochemical indicators such as TVB-N, TBARS, and biogenic amines during storage. Another research focus is the investigation of microbial succession during the storage process. This examination of microbial succession proves effective in discerning the shifts in species and quantities of dominant spoilage bacteria as they evolve during the storage period. Such insights have provided valuable guidance for the selection of preservatives and preservation methods. For example, Wei et al. (2021) conducted research on donkey meat, revealing that the dominant spoilage bacteria in this context were *Pseudomonas* spp. and *Fusobacterium* spp. According to this discovery, nanoemulsions were developed by incorporating polylysine and clove essential oil within chitosan matrices. This formulation effectively enhanced the texture attributes and extended the shelf life of the donkey meat [99,100].

The red section constitutes the foundation of the core research in this field. The selection of different preservatives and methods is based on different types of meat and different storage methods. Within this category, research is bifurcated into two main aspects: the development of preservatives on one front, and the application of physical preservation methods (such as refrigeration, freezing, vacuum packing, and air conditioning) on the other. Notably, biological preservatives with antibacterial and antioxidant properties are more favored by researchers. For instance, Liu et al. (2020) used clove essential oil to preserve Yao meat, which offered antibacterial and antioxidant effects and imparted a distinctive flavor, thus leading to a reduction in the quantity of spices required [90]. Subsequently, researchers have integrated technologies from the orange segment to jointly explore meat preservation. In these studies, various advanced techniques, including near-infrared spectroscopy [101], chromatography [102], high-throughput sequencing [103], histology [71], high-pressure homogenization [55], pulsed electric fields [104], and magnetic fields [105], are applied to assess the physical and chemical parameters of meat, microorganisms, and preservation methods and agents. These technologies have played a pivotal role in advancing the research discipline of meat storage and preservation. As depicted in Figure 7, all the components are interconnected and mutually reliant. Basic research serves as a guiding force for the advancement of research techniques, and conversely, the application of research techniques propels the deepening of the theoretical foundation.

### 3.7. Trends and Outlook of Meat Preservation

#### 3.7.1. New Packaging

Utilizing appropriate packaging methods for meat portioning, transportation, and storage can efficiently mitigate the microbial contamination of meat and safeguard against the deterioration of its physical and chemical properties [71]. Recent research on new packaging can be categorized into the following three primary areas.

##### Biodegradable, Edible Packaging

With the continuous development of the concept of green safety, public awareness of environmental protection has steadily risen. Therefore, the pursuit of novel materials that can conserve resources, safeguard the environment, and exhibit hygiene, degradability, and even edibility has become an inevitable trend in the evolution of food packaging. Biodegradable packaging is a relatively new packaging concept that has emerged in recent years. It encompasses plastic-substitute packaging created by utilizing polypropylene film [93] and biodegradable films with excellent flexibility and air-isolation characteristics, which are produced from raw materials such as starch [106], modified starch [107], cellulose [108], polylactic acid [109], and some raw materials together with plasticizers and other auxiliary materials. In parallel, there is significant research interest in edible films.

Chitosan, known for its antibacterial properties, freshness preservation, ease of film formation, and biodegradability, is widely employed in laminating and preservation films. Furthermore, the freshness-preserving capabilities are enhanced through the addition of antibacterial agents. However, compared with synthetic composite films, polysaccharide films have poor water resistance and mechanical strength [81], which could have a negative impact on meat preservation later. As a result, the future trajectory of development may involve the utilization of degradable films endowed with robust mechanical strength, exceptional water and air barrier qualities, and freshness preservation technology, in conjunction with the composite efficiency preservation approach, which incorporates antimicrobial agents into polysaccharide films as a foundational element.

While research on biodegradable edible packaging has seen some progress in the field of meat preservation, there are still several issues that need to be addressed. For instance, many polysaccharide-based films are still in the experimental stage, and their production capacity has not yet met the standards for large-scale manufacturing. And it is important to strengthen the evaluation of production capacity and economic costs for each reactive packaging technology to ensure its wide application in real-life scenarios. Meanwhile, Li et al. (2023) have pointed out that single-component films often cannot simultaneously meet requirements for mechanical performance, water solubility, and air barrier properties [110]. Therefore, in order to overcome the limitations of edible films, exploring the formulation of different components such as polysaccharides, proteins, and lipids, as well as the production processes, holds potential as a future research direction.

##### Active Antibacterial Packaging

Promising results have been observed in preserving and extending the shelf life of meat products by incorporating active substances into the packaging film matrix. Furthermore, a plethora of studies have inclined towards integrating active ingredients into the active packaging film matrix to confer antioxidant and antibacterial properties, thereby retarding the oxidative degradation of meat. In particular, the incorporation of nanoparticles (such as Ag and TiO_2_) and tocopherols, known for their antimicrobial and antioxidant attributes, into packaging materials has enhanced the antimicrobial characteristics of the packaging [57]. Additionally, the concept of attaching antimicrobial agents to the material surface through covalent or ionic bonds has opened new avenues for research. For example, Huang et al. (2019) utilized plasma treatment for covalently grafting nisin and polylysine onto PLA films, which were subsequently employed in beef preservation [111]. In summary, active antimicrobial packaging can significantly prolong the shelf life of meat by effectively inhibiting microbial growth and oxidative reactions in the long term.

##### Smart Packaging

Smart packaging is a kind of intelligent packaging with the role of extending the shelf life of products and improving product quality and safety, all while providing early warnings for potential quality issues. Meat spoilage is a consequence of the combined effects of microorganisms and environmental factors, characterized by microbial proliferation and alterations in physical and chemical parameters. The visual labeling of spoilage microorganisms, pH, and volatile odors offers a clear indication of product quality. For instance, the ToxinGuardTM packaging film can detect the presence of Salmonella and Campylobacter through an enzyme-linked reaction. When these microorganisms are present, the packaging film changes color from white to red [112]. Additionally, Kim et al. (2017) employed bromocresol green as an indicator to create a TVB-N test indicator label package for assessing the shelf life of chicken meat [113]. This highlights that the product’s condition can be visually observed through smart labels on the packaging. However, smart package labels also come with certain challenges: (1) Accuracy issues: Smart packaging labels typically indicate a single indicator, while meat spoilage results from the interplay of multiple indicators, which could potentially be misleading to consumers; (2) Safety concerns: The coloring substances used on the label may dissolve into the product itself due to moisture outflow from the meat product; (3) Economic considerations: Meat products are a common and economically significant product. The use of one or more smart labels may raise the overall price of the meat product.

Based on previous research, a recent study by Yana et al. (2023) involved the preparation of degradable smart packaging using a 10% extract derived from purple tomatoes in combination with chitosan and polyvinyl alcohol as the matrix. The results indicate that the smart packaging effectively maintains the quality of pork products and, through color changes, can monitor the decay and spoilage of the pork [114].

Zhai et al. (2020) substituted synthetic pH-sensitive dyes with natural curcumin, eliminating potential hazards. They utilized industrially produced low-density polyethylene and hot-melt extrusion technology to create smart packaging for beef preservation, achieving satisfactory results in freshness identification. Additionally, curcumin imparted functional properties, such as resistance to lipid oxidation and antimicrobial capabilities, effectively prolonging the beef’s shelf life [115].

Similarly, these studies are currently in the laboratory research phase. It is crucial to consider factors such as the potential for industrial-scale implementation, standardization of preservation stage discrimination, and how to combine them with highly isolative active films to minimize the impact of external conditions on the indicators. However, the above-mentioned research has provided us with a direction for future studies, indicating the potential to load several naturally safe indicators with monitoring and preservation functions onto low-cost, degradable films with good air barrier properties. Therefore, finding solutions to the aforementioned challenges associated with smart packaging labels and making them more practical for widespread adoption in the supermarket market will likely be a focal point for future research.

#### 3.7.2. New Preservation Technology

##### Nano-Emulsion Composite Coating Technology

The utilization of plant essential oils as both antibacterial and antioxidant agents, with their low water solubility being addressed through emulsification, has emerged as the leading edge of research in meat preservation [46]. However, the application of essential oils in food preservation still faces some limitations, such as hydrophobicity and low solubility, and the irritating odor of essential oils hinders their addition to meat in large quantities [100]. Therefore, in addition to improving the solubility of essential oils, encapsulating them in emulsion particles in a specific form provides effective stability [53]. The preparation of nanoemulsions can effectively protect food quality, slow down their volatility, and enhance controlled release, ultimately extending the shelf life of the food while reducing the negative impact of odors and off-flavors on sensory attributes. As detailed in Section 3.5 the application of nanoemulsion essential oils, which exhibit robust antibacterial and antioxidant effects alongside gradual release characteristics, in combination with coating technology, holds great promise in the realm of meat storage and preservation.

In summary, due to variations in meat types, transportation methods, storage conditions, and environmental temperatures, the release time and enhancement strategies of each type of essential oil nanoemulsion differ significantly. Therefore, in the development of nanoemulsion composite coating technology, it is essential not only to research the stability and functionality of the biobased materials themselves, but also to further investigate the characteristics and additive proportions of each component based on different application scenarios and objectives. The synergistic interaction between components in the packaging is expected to be one of the key trends in the future development of nanoemulsion composite coating technology.

##### Physical Sterilization Technology

Physical sterilization techniques help reduce dependence on chemical preservatives and mitigate the loss of quality indicators that can occur during thermal processing treatments. With the continuous development of technology, various physical sterilization methods, such as irradiation, microwaves, and pulsed electric fields, have found applications in meat preservation. Among them, emerging technologies like photodynamic inactivation and cold plasma (CP) have garnered increased attention in this field. Photodynamic inactivation (PDI) technology, characterized by its safety, efficiency, and cost-effectiveness, has been widely researched in the field of food preservation. PDI is a sterilization technology that efficiently deactivates microorganisms by stimulating photosensitizers with visible light. In practical applications, PDI technology has demonstrated effective bacterial inhibition. For instance, Li et al. (2018) utilized a 405 nm LED light source to irradiate salmon and observed a significant reduction in Listeria monocytogenes [116]. In the development of PDI technology, the current research focus lies in selecting photosensitizers (PS) that are safe, stable, and readily accepted by consumers. Studies have shown that naturally sourced curcumin and hypericin serve as excellent alternatives to traditional photosensitizers like porphyrins and their derivatives [117,118]. Lu et al. (2023) used LED as a light source to combine curcumin with citric acid to treat ground beef and refrigerate it and found that the shelf life could be extended by 2 days and the texture index could be improved, indicating that the combination of other types of preservation technology could play a synergistic role in extending the shelf life [119].

Cold plasma technology utilizes the surrounding medium of food to generate photoelectrons, ions, and reactive free-radical groups upon contact with the microbial surface, resulting in cell destruction and sterilization. The CP technology, due to its effective sterilization and low-temperature operation (60 °C), has been applied in the field of food preservation, offering excellent safety [120]. However, researchers have also found that prolonging the CP treatment time significantly reduces the a* value of red meats like pork and beef, leading to a loss of sensory quality. This is attributed to the highly oxidative nature of generated ionized oxygen molecules and free radicals [121,122].

These techniques have demonstrated effectiveness in removing bacterial biofilms from the meat surface. However, both techniques have drawbacks, including limited penetration depth and the potential to promote fat and protein oxidation. Subsequent studies can be conducted by combining antioxidants and optimizing process parameters (type of plasma gas, irradiation, angle of PID light source, device strength, treatment time) to ensure bactericidal effect and effectively reduce the loss of meat quality indexes caused by the above technology. In summary, the strategic integration of emerging preservation technologies with advanced packaging technologies will be a significant and increasingly prominent area of focus. This harmonious fusion of these technologies holds the potential to play a pivotal role in extending the shelf life of meat products, enhancing their quality and ultimately bolstering consumer satisfaction.

##### Active Functional Water

Water plays a vital role in slaughter, meat processing, and household cleaning and storage. In recent years, there has been considerable interest in the application of active functional water in the field of meat preservation. By using electrolysis, ozonization, plasma, and other technologies to activate pure water, the active water with antibacterial function is obtained. Ozone water is typically generated using low-cost electrolyzed water, and research indicates its effectiveness in eradicating foodborne pathogens. However, its 20 min half-life significantly limits practical application [123] Compared with ozone water, plasma water obtained by cold plasma discharge treatment not only has a bactericidal effect, because it is rich in NO_2_^−^ and NO_3_^−^ ions, but also has a color protection effect, which can help reduce nitrite usage to a certain extent [124].

Active functional water is easy to produce, cost-effective, and exhibits sterilization and sensory quality-enhancing characteristics, making it a promising prospect for enhancing meat quality and safety. In the future, research will likely focus on exploring the synergistic treatment effects of active functional water when combined with other technologies to effectively extend the shelf life and enhance the quality of meat.

## 4. Conclusions

This study visualized 1672 papers in the field of meat preservation using CiteSpace and VOSviewer software. Dual map visualizations illuminate the interdisciplinary nature of meat preservation research, while dynamic keyword network analyses reveal the evolving focal points of this field over time. Keyword contribution research suggests that current research hotspots focus on three aspects: (a) meat quality indicators; (b) the exploration of preservation technologies; (c) the activity and application of preservatives. Additionally, co-cited literature analysis provides a systematic overview of the knowledge underpinning meat preservation research, which was further categorized into seven collections based on intelligent software clustering to analyze and present different research categories. Moreover, the overview of different directions and technologies in meat preservation research was also obtained by summarizing the results of co-cited literature reclassification research. The highly cited literature and the most recent surges in co-cited literature emphasize the prominent role of plant essential oils with antibacterial and antioxidant attributes in ongoing research.

Furthermore, the present paper includes an outlook analysis of trends within the meat preservation field based on the literature analysis, which serves as a valuable reference for researchers seeking insights into the subject matter and its developmental direction. The pursuit of green and safe preservatives and technologies, degradable and functionally active film packaging, and visual rapid detection technology are worthy of further study. However, a single preservation technology and packaging technology have specific application scope. Thus, the rational combination of new fresh-keeping technology and packaging technology will be an emerging focus. The combination of these technologies could play a key role in extending the shelf life of meat products, improving quality, and increasing consumer acceptance. Nevertheless, the processing costs of emerging technologies and the mechanism research and safety assessment of new preservation methods should also be considered.

Compared with traditional systematic reviews, the bibliometric approach enables for a more intuitive, systematic, and scientifically grounded presentation of the discipline’s development patterns and future research directions. However, it is important to note that reliance on a single database (e.g., WOS) might lead to some limitations, such as limited data coverage, citation bias, lack of diversity, data lags, and restricted reading access for researchers. In the future, our aim is to expand our dataset to encompass more databases, thus providing a more comprehensive and three-dimensional analysis. For example, comprehensive databases (PubMed, Scopus), food field-specific literature databases (FSTA, AGRIS), patent databases, and non-English speaking country literature databases will be used to reduce the impact of a single database.

## Figures and Tables

**Figure 1 foods-12-04239-f001:**
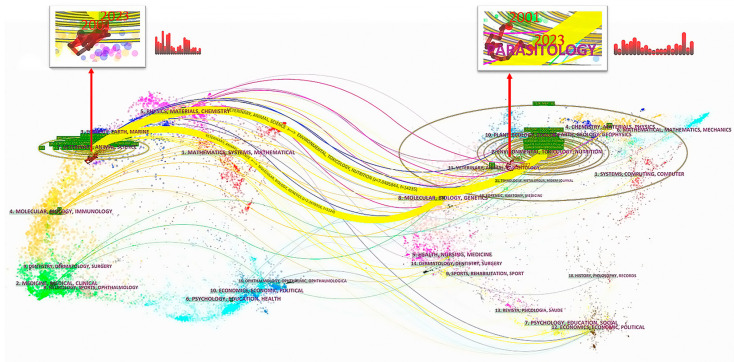
The dual map overlay of journals related to meat preservation research.

**Figure 2 foods-12-04239-f002:**
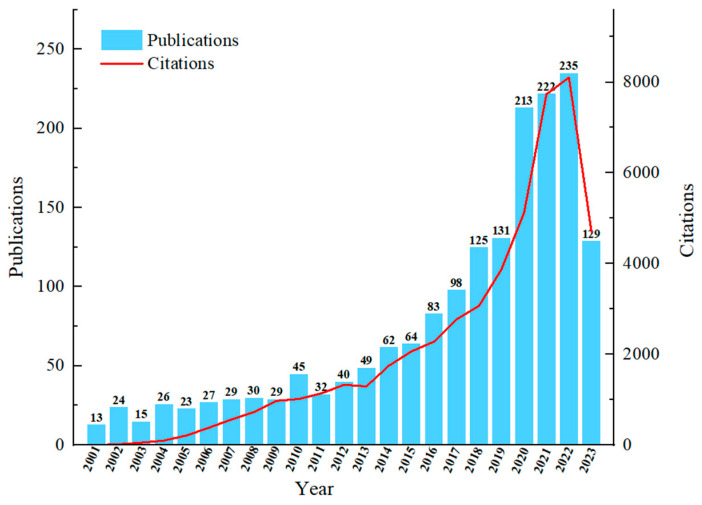
Times cited and publications over time.

**Figure 3 foods-12-04239-f003:**
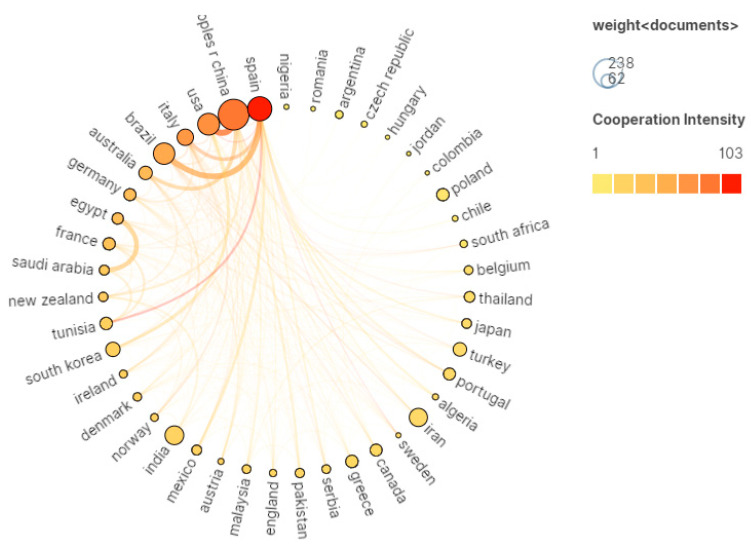
Network visualization map of cooperation between countries.

**Figure 4 foods-12-04239-f004:**
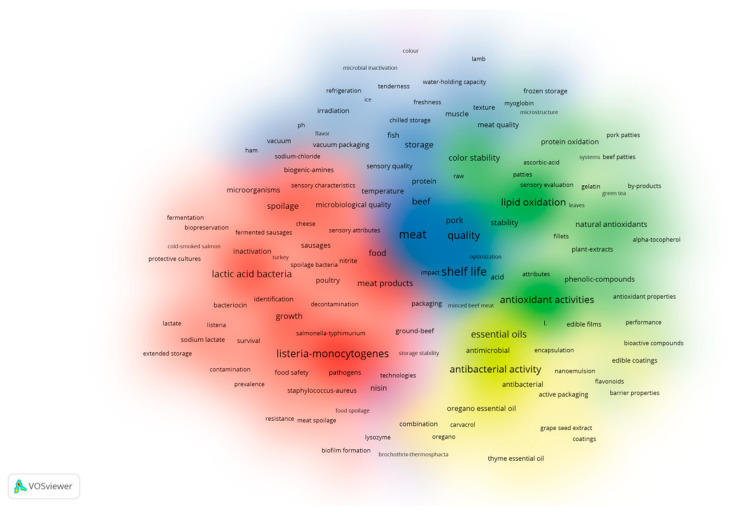
Cluster label mapping of keywords in the field of meat preservation, obtained using VOSviewer.

**Figure 5 foods-12-04239-f005:**
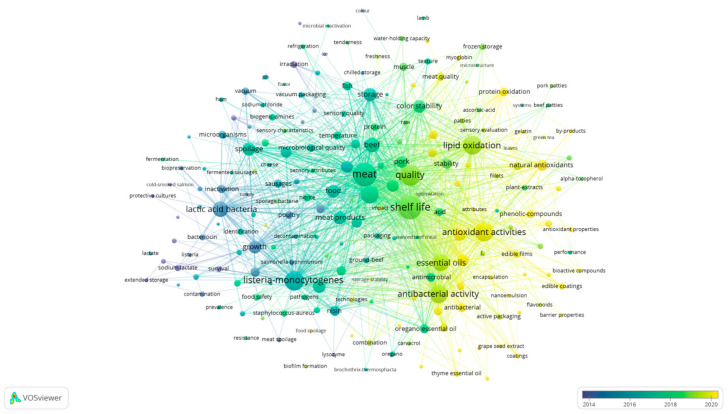
Visual mapping of keyword co-occurrence in meat preservation studiesobtained using VOSviewer.

**Figure 6 foods-12-04239-f006:**
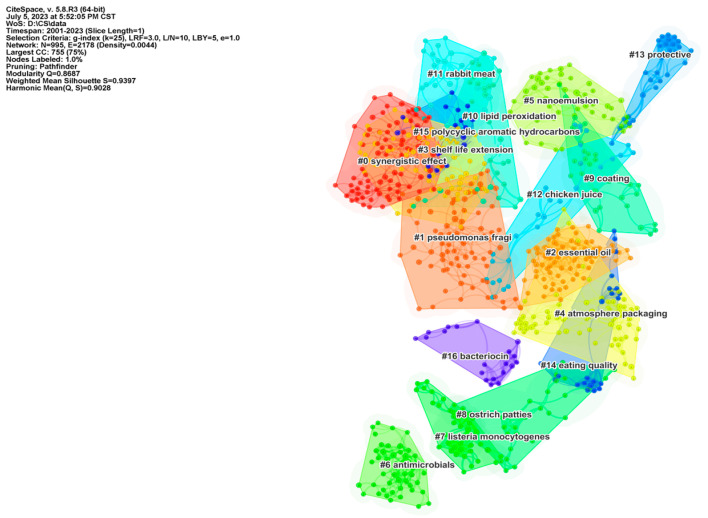
Cluster label mapping of co-cited documents in the field of meat preservation, obtained using CiteSpace.

**Figure 7 foods-12-04239-f007:**
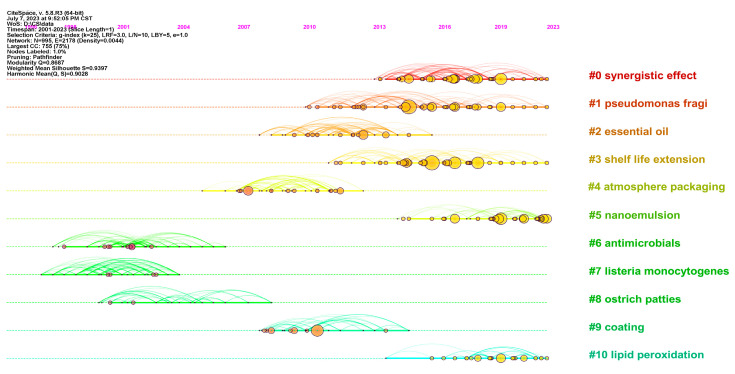
Timelines of cluster labels of co-cited documents in the field of meat preservation, obtained using CiteSpace.

**Figure 8 foods-12-04239-f008:**
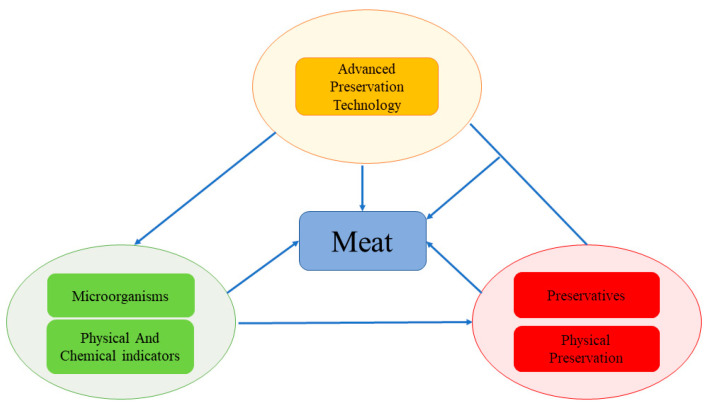
Schematic diagram of the knowledge structure framework in the field of meat preservation.

**Table 1 foods-12-04239-t001:** Top 10 countries and institutions in publications (N = number).

High Publication Countries			High Publication Institutions	
Rank	Country	N (%)	Institution	N (%)
1	Peoples R China	238 (17.02%)	Egyptian Knowledge Bank	34 (2.43%)
2	Spain	153 (10.94%)	Nanjing Agricultural University	29 (2.07%)
3	USA	124 (8.87%)	Universidade De Sao Paulo	28 (2.00%)
4	Brazil	120 (8.58%)	Spanish National Research Council	26 (1.86%)
5	India	92 (6.58%)	Indian Council of Agricultural Research	24 (1.86%)
6	Iran	87 (6.22%)	CTR Tecnol Carne Galicia	21 (1.50%)
7	Italy	70 (5.01%)	National Research Institute for Agriculture and Food	19 (1.36%)
8	South Korea	54 (3.86%)	Islamic Azad University	19 (1.36%)
9	Turkey	48 (3.43%)	United States Department of Agriculture	18 (1.29%)
10	Australia	47 (3.36%)	Universidade Estadual De Campinas	17 (1.29%)

**Table 2 foods-12-04239-t002:** Top 10 journals in publications (IF = influence factor in 2022).

High Publication Journals					
Rank	Journal	IF	Quartile	N (%)	Discipline
1	Meat Sci.	7.1	Q1	113 (8.08%)	Food Science and Technology
2	J. Food Process Pres.	2.609	Q4	85 (6.08%)	Food Science and Technology
3	LWT-Food Sci. Technol.	6	Q1	63 (4.51%)	Food Science and Technology
4	Food ControlONTROL	6	Q1	56 (4.01%)	Food Science and Technology
5	Foods	5.2	Q1	49 (3.51%)	Food Science and Technology
6	J. Food Sci.	3.9	Q2	43 (3.08%)	Food Science and Technology
7	J. Food Sci. Tech. Mys.	3.1	Q3	38 (2.72%)	Food Science and Technology
8	J. Food Process Pres.	2.609	Q4	37 (2.65%)	Food Science and Technology
9	Food Chem.	8.8	Q1	36 (2.58%)	Food Science and Technology
10	Int. J. Food Microbiol.	5.4	Q1	34 (2.43%)	Food Science and Technology

**Table 3 foods-12-04239-t003:** Highly referenced and cited literature in each cluster.

Citing Papers			Cited Papers	
Cluster	Coverage %	Author	Author	Frequency
0, 3	24	Umaraw et al. [40]	Bazargani et al. [41]	20
	16	Pateiro et al. [42]	Zhang et al. [43]	17
	15	Horita et al. [44]	Falowo et al. [45]	16
1, 7	15	Ji et al. [46]	Krishnan et al. [47]	19
	14	Zhu et al. [48]	Casaburi et al. [49]	15
	15	Gullón et al. [50]	Shah et al. [51]	14
2, 5	11	Patel et al. [52]	Noori et al. [53]	19
	10	Sánchez et al. [54]	Xiong et al. [55]	18
	9	Liu et al. [56]	Pabast et al. [57]	16
6, 16	16	Shen et al. [58]	Samelis et al. [59]	9
	15	Galvez et al. [60]	Mbandi et al. [61]	8
	13	Luchansky et al. [62]	Stekelenburg et al. [63]	8
4, 9 13, 14	17	Simonin et al. [64]	Zhou et al. [8]	21
	13	Duranton et al. [65]	Karabagias et al. [66]	11
	12	Olaoye et al. [67]	Chouliara et al. [68]	11
	12	Olaoye et al. [69]	Esmer et al. [70]	7
	12	Aymerich et al. [71]	Zhang et al. [72]	7
8, 11 12	11	Velázquez et al. [73]	Dominguez et al. [74]	22
	11	Carvalho et al. [75]	Lorenzo et al. [76]	13
	10	Alirezalu et al. [77]	Munekata et al. [46]	11
	9	Munekata et al. [78]	Ahmed et al. [79]	11
10, 15	13	Liang et al. [80]	Dominguez et al. [81]	15
	12	Bekhit et al. [82]	Fan et al. [83]	12
	11	Bekhit et al. [84]	Duran et al. [85]	10

**Table 4 foods-12-04239-t004:** Top 10 references with the strongest citation bursts.

DOI	Cluster ID	Duration in 2001–2022	Article Type	Theme
10.1016/j.meatsci.2010.04.033	9	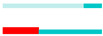	Review	Review of meat preservation technology [8]
10.1016/j.ijfoodmicro.2013.11.011	1	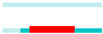	Journal article	Application of plant extracts in chicken meat [47]
10.3390/antiox8100429	11	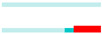	Review	Review of lipid oxidation in meat [45]
10.1016/j.foodres.2014.06.022	0	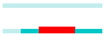	Review	Classification of natural antioxidants for meat [63]
10.1016/j.fm.2006.12.005	4	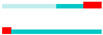	Journal article	Application of essential oil [68]
10.1016/j.foodcont.2017.08.015	5	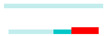	Journal article	Application of coatings based on nanoemulsions [53]
10.1016/j.lwt.2014.04.061	3	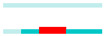	Journal article	Application of active packaging [93]
10.1016/j.tifs.2013.09.002	2	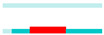	Review	Review of the effect of essential oils [56]
10.3389/fmicb.2012.00012	2	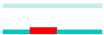	Review	Application of essential oils [94]
10.1016/j.ifset.2015.04.007	3	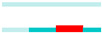	Journal article	Synergistic effect of essential [41]

(Each node from 2001 to 2022 is represented by a blue color block, and the shade of blue from light to deep indicates the frequency cited from most to least; The red marks the year of the cited outbreak.).

**Table 5 foods-12-04239-t005:** The latest references with the strongest citation bursts.

DOI	Cluster ID	Duration in 2001–2022	Article Type	Theme
10.1016/j.foodres.2018.07.014	0	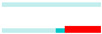	Review	Application of essential oils in meat preservation [95]
10.1371/journal.pone.0160535	0	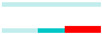	Journal article	Application of essential-oil-coated film [96]
10.3390/antiox8100429	11	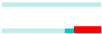	Review	Review of lipid oxidation in meat [74]
10.1016/j.foodcont.2017.08.015	5	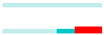	Journal article	Application of coatings based on nanoemulsions [53]
10.1016/j.foodcont.2018.03.047	5	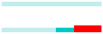	Journal article	Application of coatings based on nanoemulsions [57]
10.1016/j.meatsci.2016.03.002	0	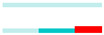	Journal article	Application of essential oils in meat preservation [97]
10.1016/j.foodres.2018.06.073	10	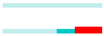	Review	Loaded essential oil films for meat applications [81]
10.1016/j.ijfoodmicro.2016.08.042	5	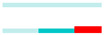	Journal article	Synergistic effect of essential oils [98]
10.1016/j.cofs.2020.03.003	11	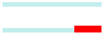	Review	Addition of plant extracts to meat [78]
10.1016/j.foodcont.2019.106771	5	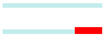	Journal article	Application of composite nanoemulsions coating [90]
10.1016/j.meatsci.2018.08.022	10	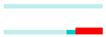	Journal article	Application of plant extracts [83]
10.1016/j.foodcont.2019.02.013	5	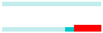	Journal article	Synergistic effect of gases and biopreservatives [89]

(Each node from 2001 to 2022 is represented by a blue color block, and the shade of blue from light to deep indicates the frequency cited from most to least; The red marks the year of the cited outbreak.).

## Data Availability

Data is contained within the article.

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
