# Peer review of "The Research Field of Meat Preservation: A Scientometric and Visualization Analysis Based on the Web of Science"

_foods, 2023, doi:10.3390/foods12234239_

Round 1

Reviewer 1 Report

Comments and Suggestions for Authors

The submitted review involves the analysis and knowledge structure mapping of a substantial number of papers related to meat preservation research, spanning over a significant period from 2001 to 2023. The study also employs various software tools, such as VOSviewer and CiteSpace, to facilitate the analysis.

The research synthesizes the knowledge structure of meat preservation research, emphasizing the relationship between basic research, technological application, and technology integration with fundamental research. This approach aligns with common principles in scientific research and knowledge development.

The study findings provide valuable assistance to researchers, helping them understand the discipline's development and identify prominent research areas.

Overall, the "Materials and Methods" section appears well-structured and follows established research practices. It provides transparency and clarity in the methodology used to conduct the study.

The authors used the Web of Science Core Collection, a reputable and widely recognized database for academic research, to ensure transparency and replicability.

As for the data selection, the authors retrieved a significant number of documents (2,331) relevant to the study's topic. They then explained that they eliminated duplicates and documents unrelated to the target topic, resulting in 1,672 documents that met their criteria. This is a common and necessary step in data curation for research studies. My concern is whether these documents should be included as supplementary of the review?

Using software tools such as Citespace, VOSviewer, Origin, Scimago Graphica, and Microsoft Excel for data analysis and visualization is entirely appropriate. These tools are widely accepted in the scientific community for conducting data-driven research and generating visual representations. The description of the specific purposes for which each software tool was used demonstrates a systematic and well-planned approach to data analysis.

The authors created various visual representations essential for summarizing and presenting complex data comprehensibly.

Author Response

Thanks for your comment. We have provided these documents as supplementary materials.

Reviewer 2 Report

Comments and Suggestions for Authors

Research field of meat preservation: a scientometric and visualization analysis based on the Web of Science

Introduction

1.      Second half of line no. 44 mentions that meat is vulnerable to spoilage during processing, transportation, and sales, leading to environmental pollution. It would be helpful to provide some examples or statistics to support this claim.

2.      The line no. 57 mentions the "clean label concept" in relation to meat preservation. It would be useful to elaborate on what the clean label concept entails and why it's important in the context of meat preservation.

Materials and Methods

3.      In the data resources and strategies in line no. 85 it was mentioned that unrelated and duplicate documents were deleted provide some insights into how duplicates and irrelevant documents were identified and excluded. This could include a brief explanation of the criteria used for this selection process.

Results and Discussion

4.      Heading 3.1 the discipline field of meat preservation research kindly start with a brief introduction explaining the purpose of the dual-map and how it contributes to the understanding of meat preservation research and also mention the source and context of the dual-map. Is it created for this study or is it from a pre-existing source? This context is essential for understanding its origin and relevance.

5.      In the heading 3.4 (Keywords of meat preservation research) explain why the keyword co-occurrence analysis is important and what understandings it can provide to researchers in the field of meat preservation.

6.      In the line 382 "pointed that" should be "pointed out that."

7.      To enhance the clarity of the passage, consider breaking it into shorter sentences or paragraphs. This can make it easier for the reader to follow the different aspects of basic research discussed.

Conclusion

8.      It's essential to acknowledge any limitations in the research process or tools used. Were there any challenges in conducting this analysis that readers should be aware of?

9.      The conclusion should concisely recap the key results and insights obtained from the analysis. What were the most noteworthy discoveries or trends in meat preservation research?

Author Response

Thank you for your letter and for the reviewers’ comments concerning our manuscript entitled “Research field of meat preservation:a scientometric and visualization analysis based on the Web of Science”. Those comments are all valuable and very helpful for revising and improving our paper, as well as the important guiding significance to our researches. Based on the reviewers’ comments, we have carefully and thoughtfully revised the paper. 

All corrections are highlighted in red in the manuscript.

Reviewer 3 Report

Comments and Suggestions for Authors

The manuscript presented biblometric evaluation of meat preservation. The manuscript is within the scope of the Journal. The English language is satisfactory, and readers will comprehend the information presented. The entire manuscript is appropriately organized. However, to meet the impact and quality of the Journal, it is important that the authors revise their work according to the following comments:

General comments:

1. Sub-sections heading: Written format/style should follow the Journal's Instructions for Authors. 

2. References list: Abbreviations of the Journal names should follow the Journal's Instructions for Authors. 

3. Spacing: Ensure appropriate spacing between text and subsections headings. Eg., Provide space at subsections 3.3, 3.4 among others.

Specific Comments

1. Introduction: This information is not adequate. There is not enough references in this section. Only 13 references were provided. A review paper should have more than 30 citations in the Introduction. Detailed background or information should be provided in this section with adequate citations.

2. Results and discussion: discussion should be Discussion

- Table 1: Check the spelling of 'Countriy'

- Table 3: What are the Frequency values for the authors Bazargani-GilaniB through Duran A.?

- The authors names provided in Table 3 are inappropriate. Eg., Umaraw, P [23] should be Umaraw et al. [23]

- Ensure that the authors' names follow the Journal's Instructions for Authors. 

- Fig. 4: Vosviewer at the end of the figure caption should be VOSviewer

- Fig. 7: Why microorganisms appeared 4 times in the Figure?

- Check the text format for Fig.7 caption 

3. Conclusions: First line....This article..should be 'This study.....

- If 1672 articles or papers were evaluated, then why only 100 references were provided in the references list? How many papers were used for this study? It is important that all papers used are provided in the list of references. The 100 references and the 1672 statement create some sort of confusion. 

Author Response

(The authors gave the same response as above.)

Reviewer 4 Report

Comments and Suggestions for Authors

The abstract provides an overview of a study analyzing research papers on meat preservation from 2001 to 2023. However, it lacks specific details about the methodology used for the analysis, making it difficult to assess the study's rigor. The abstract could benefit from more specific information about the research findings and the potential implications for the field of meat preservation.

The manuscript's introduction provides a reasonable background about the importance of meat preservation due to increased meat consumption and the potential risks of spoilage. However, it lacks a clear and specific research objective or problem statement, making it unclear what the study aims to address. Secondly, it briefly mentions the three categories of meat preservation techniques but does not delve into any details, which would have been helpful for the readers to understand the context better.

The Materials and Methods section provides some essential information about the data sources, search strategy, and software tools used in the study. However, it has a few shortcomings. First, it lacks a clear description of the data analysis process, which is critical for the reader to understand how the analysis was conducted. Second, it does not explain the criteria used to eliminate certain documents after the initial search, making it unclear why some papers were excluded. Third, while it mentions the software tools used for various visual representations, it does not explain how they were employed in the analysis or the specific methodologies used for data analysis. A more detailed and transparent description of the data analysis process and the role of each software tool would enhance the section's

While the results and discussion section provide valuable insights into the development and trends within the field, it lacks in-depth discussion and interpretation of the findings, which would help readers better understand the implications of the research trends and their practical applications in meat preservation. Additionally, it mentions challenges associated with smart packaging labels but does not delve into potential solutions or ongoing research in this area. Expanding on potential solutions or future directions would have been beneficial.

The conclusion of this article summarizes the key findings of the study, highlighting the benefits of using bibliometric analysis to understand the development and trends in the field of meat preservation. However, the conclusion lacks a critical evaluation of the study's limitations and potential biases, especially regarding the reliance on a single database, which the authors acknowledge. Additionally, the conclusion could have provided more specific and actionable insights for researchers and practitioners in the field of meat preservation, such as practical implications or recommendations for future research directions.

Author Response

(The authors gave the same response as above.)

Round 2

Reviewer 2 Report

Comments and Suggestions for Authors

All the suggestions have been incorporated

Reviewer 3 Report

Comments and Suggestions for Authors

The revised version is suitable for publication. I hereby 'Accept in present form'

Reviewer 4 Report

Comments and Suggestions for Authors

The authors have addressed the shortcomings and have substantially improved the manuscript. I am pleased to recommend the manuscript for publication in its present form.